# Identification, Structure and Characterization of *Bacillus tequilensis* Biofilm with the Use of Electrophoresis and Complementary Approaches

**DOI:** 10.3390/jcm11030722

**Published:** 2022-01-29

**Authors:** Katarzyna Pauter, Viorica Railean-Plugaru, Michał Złoch, Paweł Pomastowski, Małgorzata Szultka-Młyńska, Bogusław Buszewski

**Affiliations:** 1Department of Environmental Chemistry and Bioanalytics, Faculty of Chemistry, Nicolaus Copernicus University, Gagarina 7, 87-100 Torun, Poland; kpauter@wp.pl (K.P.); michalzloch87@gmail.com (M.Z.); mszultka@umk.pl (M.S.-M.); 2Centre for Modern Interdisciplinary Technologies, Nicolaus Copernicus University, Wilenska 4, 87-100 Torun, Poland; rviorela@yahoo.com (V.R.-P.); pomastowski.pawel@gmail.com (P.P.)

**Keywords:** antibiotics, biofilm, *Bacillus tequilensis*, capillary electrophoresis, SEM

## Abstract

Biofilm is a complex structure formed as a result of the accumulation of bacterial cell clusters on a surface, surrounded by extracellular polysaccharide substances (EPSs). Biofilm-related bacterial infections are a significant challenge for clinical treatment. Therefore, the main goal of our study was to design a complementary approach in biofilm characterization before and after the antibiotic treatment. The 16S rRNA gene sequencing allowed for the identification of *Bacillus tequilensis*, as a microbial model of the biofilm formation. Capillary electrophoresis demonstrates the capability to characterize and show the differences of the electrophoretic mobility between biofilms untreated and treated with antibiotics: amoxicillin, gentamicin and metronidazole. Electrophoretic results show the clumping phenomenon (amoxicillin and gentamicin) as a result of a significant change on the surface electric charge of the cells. The stability of the dispersion study, the molecular profile analysis, the viability of bacterial cells and the scanning morphology imaging were also investigated. The microscopic and spectrometry study pointed out the degradation/remodeling of the EPSs matrix, the inhibition of the cell wall synthesis and blocking the ribosomal protein synthesis by amoxicillin and gentamicin. However, untreated and treated bacterial cells show a high stability for the biofilm formation system. Moreover, on the basis of the type of the antibiotic treatment, the mechanism of used antibiotics in cell clumping and degradation were proposed.

## 1. Introduction

The bacterial infections associated with biofilm constitute a serious problem in the clinical area [1,2]. As previously described, biofilm impairs human defense mechanisms and creates a platform for microorganisms to use a range of strategies to control antibiotics resistance, including spore cell production, the multilayer system of bacteria cells and antimicrobial resistance [3,4].

Abiofilm is an accumulation of bacterial cells on the surface based on the agglomeration process and is surrounded by extracellular polysaccharide substances (EPSs) [5,6]. It was found that the structure of EPS shows species or strain specificity both in terms of the composition of monosaccharides, which constitute the subunit of EPS and the type of chemical bonds or non-saccharide substituents. Sometimes the form of exopolysaccharides may depend on the intensity of the cell growth. The increase in polysaccharide components in the matrix results in more free functional groups able to interact with other bacterial cells or active molecules such as antibiotics. Therefore, it determines a significant resistance of the microorganisms in biofilm [7,8,9,10,11].

Relucenti et al. [12] recommend the use of different microscopic techniques to understand the ultrastructure of the biofilm, its three-dimensional organization and the behavior of bacterial cells and their response after the antibiotic treatment. The utilization of various microscopic techniques (optic microscopy (OM); confocal laser scanning microscopy (CLSM); atomic force microscopy (AFM); scanning electron microscope (SEM)) permits imaging of the biofilm’s surface [9]. Furthermore, by combined SEM imaging with a three-dimensional image analysis system, the extracellular matrix can be quantitatively analyzed [12]. Undoubtedly, one promising analytical technique employed in the investigation of microorganisms is capillary electrophoresis (CE). This tool is also used for the separation of the microorganism’s analysis. Dziubakiewicz et al. [13] analyzed both Gram-positive (*Bacillus cereus*, *Bacillus subtilis*, *Sarcinalutea*, *Staphylococcus aureus* and *Micrococcus luteus*) and Gram-negative (*Escherichia coli*) bacteria by capillary electrophoresis. The authors pointed out that the modification of the surface charge of bacteria with Ca^2+^ allows their screening analysis with the pseudo-isotachophoretic mode of CE. The use of calcium ions allowed the number of signals on the electropherograms to be reduced, indicating a hampered ability to aggregate. Furthermore, the utilization of the pseudo-isotachophoresis mode of CE allowed the aggregates to be clustered, so the results improved. Moreover, the zeta potential was used as complementary studies [13]. In addition, Ca^2+^ was also used for the study of the yeast aggregation measurement [14]. However, in another work, Ruzicka et al. [15] demonstrated the application of capillary isoelectric focusing (CIEF) to detect also biofilm formation in *Staphylococcus epidermidis*. In this work, they report that the surface characteristics of *S. epidermidis*, based on the CIEF method, can be used to distinguish between positive and negative biofilm-forming strains [15]. The isoelectric points (pI) were between pI 2.6 and 2.3 for the strain that was able to create and not create the biofilm, respectively [15]. These discrepancies could be the result of the presence or absence of EPS on the bacterial surfaces. This crucial component of the biofilm layer enables the bacterial adhesion to solid substrates and promotes intercellular adhesion, contributing to biofilm creation. In addition, adhesion and biofilm layer formation were observed; the surface protein EPSs are involved in the antibiotic and immune system protection of bacterial cells [15]. Annet E. J. van Merode et al. [1] investigated the influence of zeta potential culture heterogeneity on retention and the biofilm creation by clinical isolates—*Enterococcus faecalis* strains. *E. faecalis* strains generally display subpopulations with various surface charges, which are expressed as zeta potential bimodal distributions. The heterogeneous strains tended to be trapped in more numbers on the polystyrene than the homogeneous strains. Likewise, the biofilm creation was much more complex for heterogeneous strains than for homogeneous strains [1]. Moreover, the matrix-assisted laser desorption ionization mass spectrometry (MALDI-TOF MS) technique was also applied to the biofilm characterization. Li et al. [16] investigated imaging of a *B. subtilis* biofilm created on agar by this technique. Thus, it was possible to visualize the distribution of metabolites of the biofilm-forming cells [16].

Therefore, the main purpose of this study was to check the usefulness of the capillary electrophoresis (CE) technique for the fast screening of the effect of different antibiotics when added on the biofilm formation using model biofilm-producing bacteria—*Bacillus tequilensis*. A variety of complementary techniques were applied here: (a) 16S rRNA gene sequencing for identification; (b) capillary electrophoresis in order to monitor the bacteria clumping of isolated bacteria in the electric field under the antibiotic treatment; (c) zeta potential measurements to check the stability of the dispersion and control the aggregation process; (d) MALDI-TOF MS on molecular profile changes; (e) fluorescence microscopy to determine the viability of bacterial cells under stress conditions; and (f) scanning electron microscopy for biofilm morphology and structure imaging. Moreover, we elucidated the role of the bacteria biofilm exposure to antibiotics belonging to different therapeutic groups, with different spectrums and mechanisms of action, as along with various chemical structures, such as amoxicillin, gentamicin and metronidazole.

## 2. Materials and Methods

### 2.1. Sample Preparation 

*Bacillus tequilensis* were chosen as a model for the biofilm formation and characterization. The isolation of the used strain was performed according to the previous protocol described by Pomastowski et al. [17]. Honey, as an accessible source, was used in the present study to isolate the *Bacillus tequilenis* strain; the isolate is a Gram-positive, non-pathogenic bacterium, which is commonly found in honey and is well known to produce biofilms. The *Bacillus* sp. are model bacterial strains for biofilm formation due to two aspects. Firstly, *Bacillus* species can produce heat-resistant endospores that play an important role in bacterial persistence and biofilm formation. Second, members of the genus *Bacillus* are known to be able to produce extracellular polymeric substances (EPS), which play a key role in the resistance of the biofilm to antibiotics by creating a mechanical barrier that restricts drug diffusion into bacterial cells. In addition, these bacteria also possess swarm motility, which can facilitate microbial survival in the environment and colonization of surfaces, leading to biofilm formation [18]. 

In this study, three different antibiotic drugs—amoxicillin (≥98% purity), gentamicin (≥98% purity) and metronidazole (≥98% purity)—were used. All of them were purchased from Sigma-Aldrich Chemicals (Madrid, Spain). The used concentrations in the present study were establish based on the minimal inhibitory concentration value according to the recommendations of the European Committee on Antimicrobial Susceptibility Testing (EUCAST) for the *Bacillus* spp. [19].

A bacterial colony was grown on Tryptic Soy Agar (Sigma-Aldrich, Steinheim, Germany) over 24 h at 37 °C. Then, the fresh cells were incubated for 2 h in Mueller Hinton Broth solution (MHB, Sigma-Aldrich, Madrid, Spain) with chosen antibiotics (amoxicillin, MIC (2 µg/mL); gentamicin, MIC (0.5 µg/mL); and metronidazole, MIC (8 µg/mL). The optical density of each suspension was adjusted according to the McFarland Standard (OD600 ≈ 3). These three antibiotics with different chemical structures (aminoglycoside, nitroimidazole, β-lactam), distinct mechanisms of action (different molecular targets—inhibition of bacterial protein synthesis, disruption of cell wall biosynthesis, interaction with the DNA) as well as various MIC values (high action—gentamicin, medium—amoxicillin and low—metronidazole) were chosen in order to change the biofilm surface structure in different ways. Three antibiotics that demonstrated various modes of action were selected, knowing that they may be characterized also by their different ability to perform EPS penetration—a key factor in determining the effectiveness of an antibiotic in biofilm eradication. Untreated cells served as a control. After the incubation step, the bacterial culture was harvested by centrifugation (13,000 rpm for 15 min, 20 °C ± 0.5 °C) and washed two times with water (5000 rpm for 5 min, 20 °C ± 0.5 °C). Washed cells were resuspended in outlet background electrolyte TB—this is a buffer solution containing a mixture of Tris base, boric acid and EDTA (Tris-borate; pH 8.0) for subsequent analysis.

### 2.2. Molecular Identification of Bacterial Biofilm Formation

The total genomic DNA was extracted using the method provided in the DNeasy UltraClean Microbial Kit (QIAGEN, Wroclaw, Poland) and was used as template for 16S rDNA region amplification using the PCR method and the universal bacterial primers 27F (5-AGAGTTTGATCMTGGCTCAG-3) and 1492R (5-GGTTACCTTGTTACGACTT-3). Obtained PCR products were sequenced using the Sanger dideoxy method (Genomed, Warsaw, Poland) and consensus sequences were compared with known reference 16S rRNA sequences found in the National Center for Biotechnology Information’s Reference RNA sequences (refseq rna) database via the Basic Local Alignment Search Tool algorithm (https://blast.ncbi.nlm.nih.gov/Blast.cgi (accessed on 3 November 2021)).

### 2.3. Capillary Zone Electrophoresis (CZE) Analysis

A CZE-UV system (Beckman Coulter, Brea, CA, USA) was used for experiments, which were performed using a fused silica capillary (length = 70 cm, inner diameter = 75 μm and outer diameter = 375 μm, Beckman Coulter) with an effective length of 50 cm. New capillaries were rinsed prior with Milli-Q water, 1 M NaOH, 0.1 M NaOH, Milli-Q water and a running buffer, each for 2 min. Before each day, the capillary was treated with 1 M NaOH followed by distilled water for 30 min. To ensure repeatability, between each analysis, the capillary was rinsed with the background solution for 30 min. At the separations, an inlet buffer—TBH (Tris-borate, hydrochloric acid; pH 7.3)—and an outlet buffer—TB (pH 8.0)—were used. The buffer solutions were prepared by dissolving the appropriate amount of H_3_BO_3_ and Trizma^®^ base in deionized water and the pH was adjusted with 0.1 M HCl (Sigma Aldrich, St. Louis, MI, USA) in case of TBH, and 0.1 M NaOH (Sigma Aldrich) in case of TB. The boric acid came from Chempur (PiekaryŚląskie, Poland). The Trometamol (Trizma^®^ base) was purchased from (Sigma-Aldrich, USA). Each sample was injected into the capillary at 8 psi for 10 s. UV detection wavelength was set at λ = 214 nm and the capillary temperature was maintained at 23 °C. The separation procedure was adapted based on the research of Dziubakiewicz et al. [13]. Electropherograms included migration time and the peak area; they were recorded and processed using the 32 Karat v.8.0 software (Beckman Coulter, Brea, CA, USA). Fractions for the SEM analysis were collected directly into the outlet vial during the CE analysis. Thiourea was used as a determinant of electroosmotic flow (EOF), (Sigma-Aldrich, Bangalore, India).

### 2.4. Zeta Potential Assay

For the zeta potential of bacterial cell measurements, Zetasizer Nano ZS was used (Malvern Instruments, Malvern, UK). The bacterial suspensions in TB buffer were prepared according to the method published by Buszewski et al. [20]. Additionally, the samples were vortexed for 15 min at 22 °C, and subsequently sonicated (5 min, 25 °C). The values of the zeta potential were summarized as the average value based on the three measurements.

### 2.5. MALDI-TOF MS Analysis 

MALDI-TOF was used to analyze the molecular profile of *B. tequilensis*. Protein extracts were obtained from bacterial cells using a quick method described by Pomastowski and co-workers [17]. To the bacterial pellets, 900 µL of ethanol was added, then centrifuged at 14,400 rpm for 5 min at 20 °C. The supernatant was removed and the pellet was dried using vacuum concentrator for 8–10 min. 

The material was then dissolved in 10–12 μL of 70% formic acid (Merck, 98–100%, Darmstadt, Germany), which was followed by the addition of the same volume of acetonitrile (Fluka Analytical Sigma-Aldrich, Munich, Germany). The resulting mixture was centrifuged again at 13,000 rpm for 2 min. Finally, 1 μL of each control bacteria and bacterial suspension with antibiotics were transferred to one sample spot on a MALDI target plate and allowed to dry at room temperature before being overlaid with 1 μL of HCCA matrix solution (10 mg/mL in 47.5% water, 50% ACN, 2.5% TFA), the calibration on BTS being in the quadratic mode. The visual evaluation of the mass spectra was carried out initially with the Flex Analysis 2.4 software (Bruker Daltonik GmbH, Bremen, Germany). Then, the raw spectra were processed using the MALDI BioTyper 1.1 software (Bruker Daltonik GmbH). The obtained raw data have been additionally processed by STATISTICAL Release 7 software in order to identify the signal changes on the protein profile of untreated and treated bacteria cells. The data were visualized by the Scatterplots w/histograms model.

### 2.6. Fluorescence Microscopy Assay

The effect of a bacterial viability under the antibiotics treatment was investigated through the fluorescence microscopy. The bacterial cells were stained regarding the method reported previously by our group using the ethidium bromide and acridine orange as a dyes in order to differentiate the live and dead cells to be able to establish the level of the modified biofilm structure [21]. For each sample, fluorescence images were taken with a Zeiss Axio Observer.D1 (Zeiss, Oberkochen, Germany) by the Axio Vision 4.8. software (Zeiss, Oberkochen, Germany). 

### 2.7. Scanning Electron Microscopy Assay

Scanning electron microscopy (SEM) was performed to add a visual aspect of the effects of used antibiotics on the biofilm morphology and structure. The samples were prepared in simple way by applying a small drop of bacteria cells suspension on the dedicate microscope slide. Then, the specimens were completely dried and directly examined by the scanning electron microscopy (SEM) (SEM/FIB—Quanta 3D FEG, FEI, Gräfelfing, Germany). Moreover, this technique was also applied to observe the changes in biofilm structure after the CZE-UV technique. In addition, the changes in biofilm surface were also pictured for the collected fractions using the capillary electrophoresis technique. The fractions were collected based on all recorded signals on the electropherogram. 

## 3. Results

### 3.1. Molecular Identification of Bacteria Biofilm Formation

The identification of the bacterial isolates was carried out by the polymerase chain reaction (PCR) and sequencing of a conserved 16S rRNA gene fragment. For the PCR reaction, two commonly used primers, 27F and 1492R, were used. Regarding the PCR method, isolated bacteria was identified at 99.78%, as a Gram-positive *B. tequilensis* strain. The evolutionary timeline of the study bacteria was presented by constructing a phylogenetic tree (Figure 1). 

We determined the evolutionary history based on the Neighbor-Joining method [22]. For the percentage of replication trees in which related taxa clustered along in the bootstrap test (500 replicates), it is presented next to the branch [23]. The tree is plotted to scale, with the branch lengths being of the same units as the evolutionary distances used to draw the phylogenetic tree inference. Therefore, the evolutionary distances were calculated according to the p-distance method [24] and given in units of the number of baseline differences for each position. The data evaluation was performed in the program MEGA7. It was observed that *B. tequilensis* next to *Bacillus subtilis* created the biofilm after 12 h of incubation, through the creation of endospores [18]. *Bacillus* sp. are model bacterial strain to the biofilm formation [25].

### 3.2. CZE Assay

After the molecular identification, the migration of untreated and treated bacteria cells with the chosen antibiotics was performed via the CZE-UV technique. Bacterial cells yielded signals with migration times longer than thiourea (2.58 min), confirming the negative bacterial charge. On the basis of the electropherograms (Figure 2), it can be observed that the intensity of the signals depends on the used antibiotic. In case of amoxicillin and gentamicin, the intensity of the signals was significantly lower compared to control. The large number of signals in the case of control may indicate a natural tendency to form aggregates. The sum of the surface area signal for the control was 100% (2,170,436), 14.75% (320,311) for amoxicillin, 19.67% (426,867) for gentamicin and 125.89% (2,732,344) for metronidazole. This may indicate that the remaining bacterial cells create the resistance effect. Moreover, the electropherogram of the bacterial cells treated with metronidazole was observed to be closed to the control sample. What is more, after the addition of the antibiotics, the disappearance of the last signals was registered at 12.10 ± 0.03, 12.23 ± 0.03 and 12.42 ± 0.05 min with 87.40, 23.25 and 6.45 mAU, respectively, compared to the control. This aspect indicates the presence of the clumping phenomenon (disappearance of large aggregates) (Figure 2). Based on Figure 2, it can be seen that the signal at 4.22 min in the control had the highest intensity and area (231,793).

However, when the control was treated with antibiotics, the signals had a significantly lower area (amoxicillin 6152, gentamicin 15,171) and intensity. In the case of metronidazole, it can be seen that the signal was divided into two signals with the surface area signal equal to 88,124 and 63,942. In addition, in case of metronidazole, the surface area signal was noticed to be less compared to the control.

It was reported by many authors that this technique can be a promising screening tool to control the changes in bacterial cell membranes after antibiotic treatment and it may predict the mechanism of their action. Kłodzińska et al. [26] demonstrated the interactions of antibiotics with bacteria isolated from patients with postoperative wound infections by capillary electrophoresis. It was noticed that, in case of the sample after the antibiotic administration, the spectrum generated by the *Escherichia coli* cells becomes diffuse and broad. This indicates the interaction of the antibiotic (gentamicin) with the bacterial cells. Moreover, this study elucidates the action mechanism of antibiotics on *E. coli* cells, suggesting changes in cell membrane permeability and the slow lysis of cells [26].

Indeed, one of the conditions for the formation of a monolayer of bacterial cells on a given surface is the overcoming of electrostatic repulsive forces by hydrophobic interactions of an attractive nature. According to the literature data [26,27], bacterial cells are usually endowed with a negative surface electric charge, the value of which depends on, among others from the strain, chemical structure and surface of the cell. The reason for charge formation in the bacterial cell wall was the dissociation of the surface group of the molecular component of cell walls (peptidoglycan and proteins). 

The surface electric charge of cells influences the rate of their movement in an electric field and also the value of their electrokinetic potential, which characterizes the electric double layer around the bacterial cell. In our study, the applied capillary electrophoresis allowed us to understand the changes occurring in the cell membrane of biofilm-forming bacteria under exposure to antibiotics.

### 3.3. Zeta Potential Analysis

CE was reported to constitute one of the techniques involved in the biofilm characterization. However, it is necessary to underline that each technique has its limitations. In fact, only the combination of CE with another approach can give complimentary characteristics [13,27,28,29]. Therefore, in the present study, the zeta potential analysis was performed in order to elucidate the impact of various antibiotics with different action spectra on the biofilm strain. For the sake of comparison, the untreated cells were also investigated as a negative control (absolute value). For untreated cells the ZP value was found to be ξ = −43.65 ± 1.06 mV while for treated cells ξ = −51.50 ± 1.44 mV for amoxicillin (BT-AMOX), ξ = −48.35 ± 0.92 mV for gentamicin (BT-GEN) and ξ = −55.50 ± 1.56 mV for metronidazole (BT-MET). As is well known, the optimal value determining the stability of the dispersion is around ±30 mV [27]. Therefore, since the established zeta potential value (ξ = −43.65 mV) for the untreated bacterial cells is more than −30 mV, such a system is considered stable [27]. Furthermore, it was noticed that the ZP value of bacteria cells treated with applied antibiotics caused an increase in the dispersive stability of the system, which may be caused by the increased impact of solvation resulting in the degradation of bacterial cells into smaller micellar systems. 

It was demonstrated that the exposure to antibiotics can cause changes in bacterial surface characteristics. Thornsberry et al. reported [30] that β-lactams inhibit the stabilization of new cell-wall components. It was recognized that antibiotics can cause changes in zeta potential value before any signs of cell death appear [30]. Furthermore, Kłodzińska et al. [26,27] suggest that the zeta potential measurements are useful in order to understand the separation electrophoretic behavior and provide more information about the charge that is present on the surface of the bacteria cells [26,27].

The presence of an electrical charge on the surface of microorganisms consequently has a direct impact on aggregation and adhesion to solid surfaces, and can also serve as a parameter to distinguish between different bacterial strains. The electrical properties of bacterial cells can therefore be characterized by measuring the zeta potential, as one of the electrokinetic parameters related to dispersion stability and solvation phenomena, which is the electrical potential generated between the solution and the solid layer (capillary surface, stationary phase) [1,27].

We noticed that the value of zeta potential increases; therefore, the electrostatic interactions between bacterial cells and antibiotics will be stronger, the stability of the systems will increase and the size distribution will be more homogeneous. All the zeta potential values found in this study were negative and relatively high, indicating good physicochemical stability of colloidal suspensions.

### 3.4. MALDI-TOF MS Analysis 

The MALDI-TOF MS technique was performed to follow the molecular level changes of the untreated bacteria cells and cells affected by the antibiotic treatment. The UNIPROT database was used to associate the selected signals with the corresponding protein names and their function. The mass spectra (Figure 3A) presents the common signals recorded at 2882 and 6934 m/z in case of all investigated samples. 

The protein profile and intensity of these signals are dependent on the applied antibiotic. One can also observe three signals (m/z at 5190, 6187, 6435) occurring in case of treated and untreated bacteria cells that belong to sporulation protein [18,19,20]. Remarkably, for BT-AMOX and BT-GEN, the characteristic protein profile was noticed (Figure 3B). The typical changes appeared at 3711, 5248 and 5294 and 9722 m/z. According to the UNIPROT database, the protein responsible for the formulation of cellular spores was identified (3711 and 5248 m/z). It constitutes a structural component of the ribosome (5248 m/z), transcription (5294 m/z) and RNA binding (9722 m/z) [31,32]. The MALDI spectra recorded in the case of the cells treated with metronidazole is the most similar to the control, based on the weak action and mechanism of action involving stimulation of production of DNA-destroying compounds of microorganisms inside cells (Figure 3C). The signals at 6720, 9552 and 9878 m/z are responsible for the constitution of the transcriptional, regulation and proper spore morphogenesis (one of the formation mechanisms of biofilm), respectively, of the ribosome [32,33]. Additionally, the disappearance of some signals (50 ribosomal protein (6720 m/z), in the spore morphogenesis (9878 m/z) was noticed [32,33]. In the case of the biofilm treatment with amoxicillin and gentamicin, the strong effect of these antibiotics and their mechanisms of action were indicated. Amoxicillin is responsible for inhibiting cell wall synthesis while gentamicin blocks the ribosomal protein synthesis.

In the case of metronidazole, the molecular profile was the most similar to the control, indicating a limited entry of antibiotic molecules into the cells and, therefore, missing access to their molecular target, which is the microbial DNA [31,32,33].

Figure 4 presents a comparison between the protein profiles of *B. tequilensis* cells untreated and treated with antibiotics (Figure 4A–D), representing a comparison of signals appearing noticed on MALDI-TOF mass spectra for control *Bacillus tequilensis* cells (x axis) and under antibiotics treatment (y axis) which were proteins (marked with different numbers #1–#10) identified using UNIPROT data first of all, with base proteins being listed in the figure captions with their m/z values. The recorded signals were clustered together mainly in three clusters. In the case of cells treated with metronidazole, the clustering way was close to the control samples (Figure 4A,B).

This indicates that metronidazole displays weak profile changes. In turn, in the case of the samples treated with amoxicillin and gentamicin, the clustering takes place differently; the profile clustering was very similar in those cases (Figure 4C,D), which suggested a similar effect on the proteomic profile of the *B. tequilensis*. Moreover, the cluster groups are different compared to the control. First of all, the downregulation of the proteins associated with the regulation of the protein translation and transcription along with the proper structure of the ribosomes in amoxicillin and gentamicin treated cells were noted: 6720 m/z—50S ribosomal protein L32 (structural constituent of ribosome; Uniprot, [32]), 9552 m/z—Protein Veg (regulation of transcription, DNA-templated; Uniprot, [33]). Furthermore, MS profiles of the treated cells lacked signals derived from the protein YwcE required for proper spore germination: disturbance of the spore production is mainly marked by a lack of the signal 9878 m/z in the *B. tequilensis* cells treated with gentamicin and amoxicillin.

SEM images from amoxicillin and gentamicin variants supported these findings, since disrupted cells or cells with distorted cell wall morphology were not accompanied by the presence of spore-like structures, which should appear after cell envelope destruction when the sporulation process is not disturbed. Interestingly, the addition of the metronidazole failed in the expression of the mentioned cellular components, although its mode of action is that it, when diffused into the organism, inhibits protein synthesis by interacting with the DNA. Such observations suggested different biofilm penetration abilities by amoxicillin and gentamicin compared to metronidazole; namely, the latter failed to reach the molecular target inside a cell in sufficient amounts to reveal biological activity against *B. tequilensis*.

Pereira et al. [34] evaluated the MALDI-TOF mass spectrometry to analyze the molecular profile of *Pseudomonas aeruginosa* biofilms grown on glass and plastic surfaces at different stages of biofilm development. Results from molecular studies showed that profiling based on MALDI is not able to distinguish between different stages of biofilm development, but it may be observed when biofilm cells are released in the dispersion stage, which occurred first on the polypropylene surface. Furthermore, the present study indicates that MALDI profiling may become a promising technique for a clinical diagnosis and prediction of the biofilm formation development [34]. Additionally, Si et al. studied [35] the molecular heterogeneity in *B. subtilis* colony biofilms by using the MALDI-TOF MS. In this work, they combined the MALDI and fluorescence method, which permitted the detection of distinct cell populations in the biofilm [35]. In our study, we demonstrated that matrix-assisted laser desorption time-of-flight mass spectrometry with the UNIPROT database can be used as a complimentary technique to CE to study differences in the molecular profile of *B. tequilensis*, after antibiotic treatment. Similarities of molecular profiles of control and bacteria treated with metronidazole were observed; the same was registered in electropherograms of the bacteria strain with antibiotics.

### 3.5. Fluorescence Microscopy

Fluorescence microscopy was performed in order to supplement the hitherto obtained observations concerning the investigation of the viability of bacterial cells under antibiotic treatment. Figure 5 depicts the fluorescence images of the untreated biofilm formed by *B. tequilensis* bacteria and treated with antibiotics. Amoxicillin and gentamicin showed reddish-yellow colored areas, indicating the presence of damage in the bacterial cells.

In addition, a significant destruction of the surface of the biofilm formation was observed, especially with gentamicin. Moreover, after the metronidazole treatment, a slight surface changes of the biofilm can be noticed. This aspect indicate the low effectiveness of the respective antibiotics. The impact of antibiotic on *B. tequilensis* (control) formation was as follows GEN > AMOX > MET.

Amoxicillin and gentamycin are the strong bactericidal antibiotics. According to the European Committee on Antimicrobial Susceptibility Testing (EUCAST), their minimum inhibitory concentration (MIC) is 2 µg/mL for amoxicillin and 0.5 µg/mL for gentamicin [19]. Therefore, both antibiotics induced bacteria biofilm damages and even bacterial cells damages of *B. tequilensis*. Notably, metronidazole had a measurable effect on the biofilm viability. However, this could be associated with its weak bactericidal activity (MIC = 8 µg/mL) [19].

On the other hand, earlier research suggested that antibiotics move through biofilms quite rapidly. However, Tseng et al. [36] analyzed the penetration of two clinically relevant antibiotics, tobramycin (aminoglycoside) and ciprofloxacin (fluoroquinolone), into *Pseudomonas aeruginosa* biofilms by using confocal fluorescence microscopy. They demonstrated that the positively charged antibiotic tobramycin is sequestered at the biofilm periphery, whereas the neutral antibiotic ciprofloxacin readily penetrates. Furthermore, they evidenced that tobramycin applied at the biofilm periphery both stimulated a localized stress response and caused the death of bacteria cells those regions, but not in the deeper layers of the biofilm [36]. In our study, a similar observation was noticed in reference to *B. tequilensis* cells treated with gentamicin. Except for the efficient mechanism action on gentamicin, the surface bacterial strain also was sensitive to the respective antibiotic, in contrast to the deeper layer as was noticed on the SEM results before and after electrophoresis (Figure 6 and Figure 7).

Oubekka et al. [37] studied the influence of vancomycin for the biofilm created by *S. aureus* human isolates by the advanced fluorescence microscopy method (fluorescence recovery after photobleaching (FRAP), fluorescence correlation spectroscopy (FCS) and fluorescence lifetime imaging (FLIM)). They demonstrated that at therapeutic concentrations of vancomycin, the biofilm matrix was not an obstacle to the diffusion reaction of the antibiotic, which can reach all cells through the biostructure [37].

To conclude, we can suggest that the bacterial cells in the deeper layers of the biofilm have the ability to adapt to the environment and acquire resistance mechanisms as a consequence of molecular mechanisms. It can be assumed that the results are correlated with the findings received by using the MALDI-TOF MS technique and CE.

### 3.6. Scanning Electron Microscopy

In order to check the changes on the surface and morphology of the biofilm under the influence of tested antibiotics, scanning electron microscopy (SEM) imaging was applied. Moreover, such technique was used as a complimentary approach to prove the capillary electrophoresis results and to monitor how different antibiotics with different action mechanisms can influence the migration of the investigated bacterial cell in an electric field.

Figure 6 and Figure 7 present the SEM images of the biofilm formation of *B. tequilensis* bacteria treated with antibiotics, before (Figure 6) and after the capillary electrophoresis analysis (Figure 7). On the basis of the SEM results, before and after electrophoresis, two aspects were observed: the changes on the surface area of the biofilm and the inhibition of the bacteria growing. As can be seen in Figure 6 and Figure 7, the strong effect was observed for GEN and AMOX, compared to the control. Nonetheless, the effect of metronidazole was slightly similar to the untreated cells. Gentamicin significantly destroyed the bacterial cells. On the other hand, after the metronidazole treatment, the change was mainly in the biofilm surface.

Bacterial cells treated with metronidazole were less affected on the surface of the biofilm compared to the amoxicillin and gentamicin treatment. The shape of biofilm-forming cells was also found to be significantly modified depending on the antibiotic used.

After the metronidazole treatment, cells adhered to the surface were more elongated and had a narrow size compared to the amoxicillin and gentamicin exposed, whereas the bacterial cells of BT-AMOX and BT-GEN had a similar oval shape.

Remarkably, the different correlations in the change of the biofilm area and morphology in the case of treated cells were observed after the capillary electrophoresis (CZE-UV) analysis (Figure 7).

Notably, following the CZE-UV, clustering was noticed in both the treated and untreated cells of the biofilm. They created complex structures with different shapes. However, in the case of metronidazole, besides the overlapping cells with a similar pillow shape, the presence of branching, which is also present in the control, was also noticed. It was observed that bacterial cells treated with gentamicin formed more clusters than those treated with amoxicillin.

The SEM images of biofilms treated with amoxicillin and gentamicin displayed substantial changes to the biofilm structure—fewer cells and larger areas without cells. Thus, both of these antibiotics kill bacteria and clear large sections of the biofilm, thereby penetrating inside the structure to exert their bactericidal effects. The images obtained seem to reflect the proteomic changes noted during the MALDI analysis—mostly related to the disturbance of proper spore production. Indeed, a similar observation was noted in the work by James et al. 2018 [38] for the *Clostridium difficile* biofilm treated with surotomycin and fidaxomicin, where both vegetative cells and spores were killed by the aforementioned antibiotics, which were reflected in different biofilm images than the control one, thus negating the protective role of the biofilm structure in that case. Moreover, the obtained images reflected mechanisms of action under the influence of both antibiotics—amoxicillin disturbs proper cell wall formation, which prevents the proper process of cell division (whole cells but deformed morphology of the cell wall), while gentamicin deregulates the bacterial protein synthesis by irreversibly binding to 30S ribosomes and inducing a significant increase in the misreading of messenger RNA (disintegrated cells). It is also significant that one important structural feature of biofilms is that the bacteria are embedded in a self-reproducing EPS matrix. These proteins include polymers that impede the transport of antibiotics into the biofilm. Hence, the biofilm matrix can provide a protective barrier by neutralizing antibiotics [8]. However, depending on the antibiotic treatment, their penetration through the matrix is different. Andert et al. [39] studied the penetration of ciprofloxacin and ampicillin for the biofilm created by *Klebsiella pneumoniae*. They noticed that ampicillin was unable to penetrate and was captured by high concentrations of β-lactamases secreted by bacteria living in the biofilm [39]. Since amoxicillin demonstrated a visible destroying effect on the investigated *B. tequilensis* biofilm, the accumulation of antibiotic-degrading enzymes such as β-lactamases in the biofilm matrix as a defense mechanism should be excluded. Contrary to this, the penetration of the biofilm through the metronidazole seems to be limited—biofilm morphology was similar to the control. Metronidazole inhibits the bacterial protein synthesis by interacting with the DNA, thus, such an observation may suggest the presence of the extracellular DNA (eDNA) in the biofilm matrix produced by investigated *B. tequilensis* strains that interact with metronidazole and prevent it entering cells. eDNA is a significant and common component ingredient of the bacterial biofilm matrix that can increase the biofilm’s resistance to certain antimicrobial agents [40].

Through this technique in our study, changes in the surface layers of the biofilm could be seen depending on the antibiotic used both before and after capillary electrophoresis (Figure 6 and Figure 7). The changes in the surface of the bacteria cells correlate with the results in the deeper layers of the biofilm.

On the basis of the findings achieved by capillary electrophoresis, the MALDI-TOF MS technique and microscopic studies, we proposed the mechanism of action of the studied antibiotics (amoxicillin, gentamicin, metronidazole) on the biofilm formation (Figure 8).

The proposed scheme is concerned with the mechanism of studied antibiotics on the phenomenon of clumping and degradation in the bacterial cell of *B. tequilensis*. According to MALDI profiles, the molecular changes induced by the used antibiotic and identified by the UNIPROT database are related to aggregates formation, visible by CE. The creation of aggregates results in changes in the charge measured by the ZP approach. More destroyed or sensitive bacterial cells creates a more dispersive system and less aggregation in fused silica under the CE analysis. The main mechanism included: the degradation of cell wall synthesis; the disturbance of the cell membrane; the inhibition of the DNA and ribosomal proteins; and the dysfunction of transcription. Regarding observed data that can decipher potential molecular mechanisms involved in biofilm development disturbance, both SEM images, CE analysis as well as results of the proteins’ profile changes indicated significant differences between investigated antibiotics in terms of cell morphology, presence or lack of specific proteins related to spore production, protein transcription and regulation, reflected by different electrophoregram images.

## 4. Conclusions

This study is the first complex and complementary approach in the characterization of biofilm both before and after antibiotic therapy. Owing to the 16S rRNA gene sequencing method, the bacteria isolated from honey was identified as a model of biofilm formation—*Bacillus tequilensis*.

Remarkably, capillary electrophoresis was found to be a useful tool in biofilm characterization. The application of this technique enabled us to observe changes in the electrophoretic mobility of bacterial cells untreated and treated with antibiotics (amoxicillin, gentamicin, metronidazole). It was noted that the treatment with amoxicillin and gentamicin caused the disappearance of large aggregates (clumping phenomenon). In this case, there were strong changes in the electrical charge on the cell surface. Furthermore, the use of complementary techniques allowed us to monitor differences in dispersion stability and molecular profiles, along with the viability and morphology of bacterial cells creating the biofilm when exposed to antibiotics.

Once the zeta potential increases, therefore, the electrostatic interactions between bacterial cells and antibiotics will be stronger, the stability of the systems will increase and the size distribution will be more homogeneous.

The increasing of the zeta potential value indicates that strong electrostatic interactions between bacterial cells and applied antibiotics were observed. The treated and untreated bacterial suspensions showed negative zeta potential values, which is related to a good stability of colloidal systems.

The MALDI-TOF MS analysis indicated changes in the molecular profiles of *B. tequilensis* before and after the antibiotic therapy, leading to proposed mechanisms of antibiotic resistance. However, the use of fluorescence microscopy showed that bacterial cells in the deeper layers of the biofilm are able to adapt to the environment and acquire resistance mechanisms as a result of molecular mechanisms. Finally, scanning electron microscopy was used to observe changes in the biofilm surface under the influence of antibiotics both before and after capillary electrophoresis. According to microscopic and mass spectrometric analyses, it was noticed that amoxicillin and gentamicin caused the degradation of the cell wall synthesis, the disturbance of the matrix of extracellular polysaccharide substances surrounding the biofilm (EPSs) and the inhibition of ribosomal proteins and the dysfunction of transcription. This phenomenon is correlated also with the noticed higher absolute value of ZP and less registered signals on CE. It should be pointed out that the results obtained by MALDI-TOF MS, fluorescence microscopy and SEM were corelated with the capillary electrophoresis method. Therefore, in this study, we proposed a mechanism of examined antibiotics (amoxicillin, gentamicin, metronidazole) on biofilm formation, associated with the phenomenon of clumping and degradation, which may facilitate the treatment of bacterial infections related to biofilm.

## Figures and Tables

**Figure 1 jcm-11-00722-f001:**
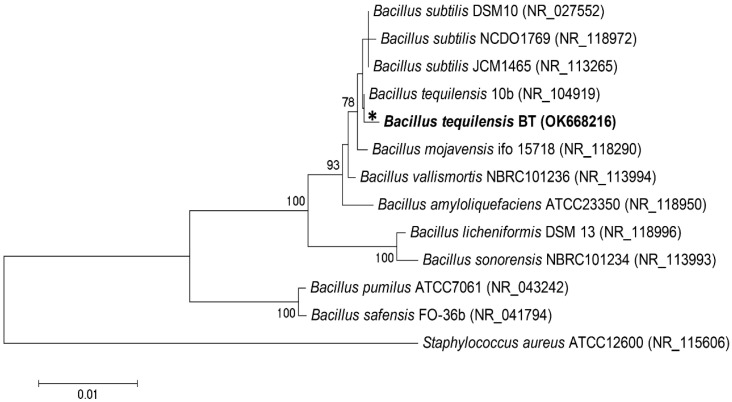
The evolutionary timeline of the study bacteria as a phylogenetic tree. *—indicate the isolated strain and used in present study.

**Figure 2 jcm-11-00722-f002:**
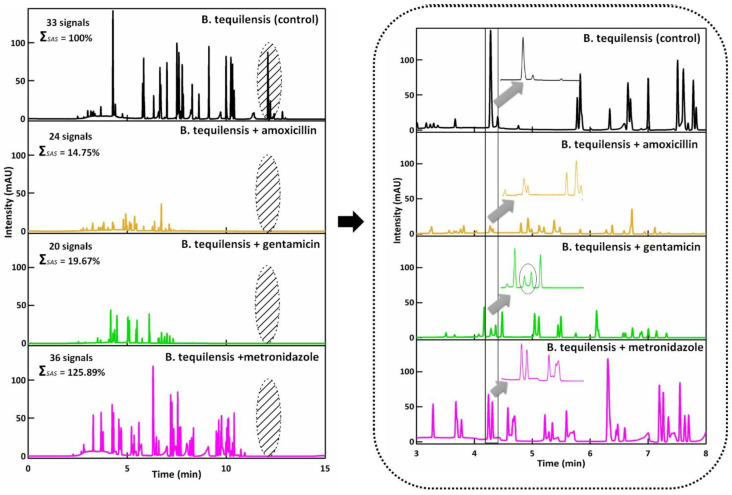
Electropherograms of the *B. tequilensis* treated antibiotics. Electrophoretic conditions: inlet buffer TB (pH 7.3), outlet buffer TBH (pH 8.0), U—20 kV, T—25 °C, injection—8 psi/10 s, λ = 214 nm, capillary—L_tot_ = 70 cm; L_eff_ = 50 cm; and i.d. 75 μm. ΣSAS—sum of surface area signal.

**Figure 3 jcm-11-00722-f003:**
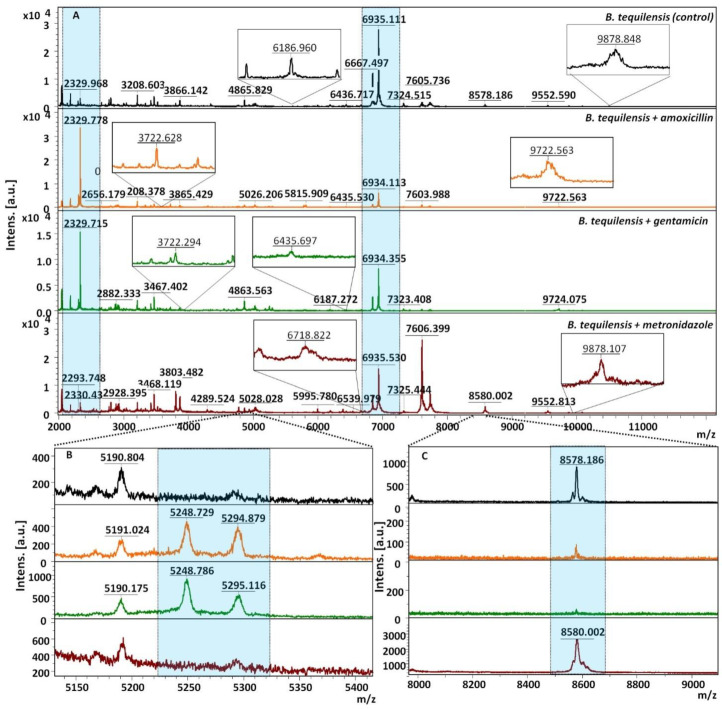
MALDI-TOF MS mass spectra of bacteria—*Bacillus tequilenis* after influence of antibiotic drugs. (**A**) main MS spectra, (**B**,**C**) represent the zooms MS spectra.

**Figure 4 jcm-11-00722-f004:**
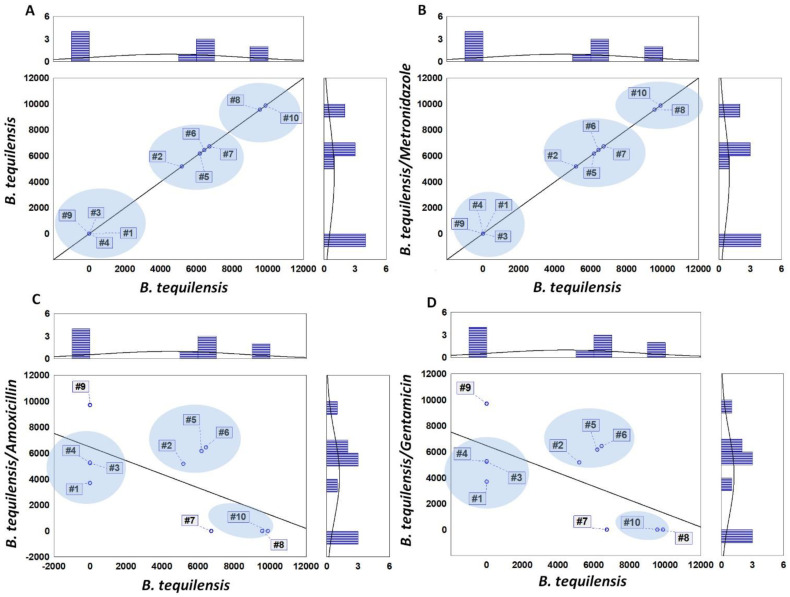
Comparison of signals noticed on MALDI-TOF mass spectra for *Bacillus tequilensis* (control) and under antibiotics treatment: (**A**) control, (**B**) *B. tequilensis*/metronidazole, (**C**) *B. tequilensis*/amoxicillin, (**D**) *B. tequilensis*/gentamicin. #1—3711 m/z, #2—5190 m/z, #3—5248, #4—5294 m/z, #5—6184 m/z, #6—6435 m/z, #7–6720 m/z, #8—9552 m/z, #9—9722 m/z, #10—9878 m/z.

**Figure 5 jcm-11-00722-f005:**
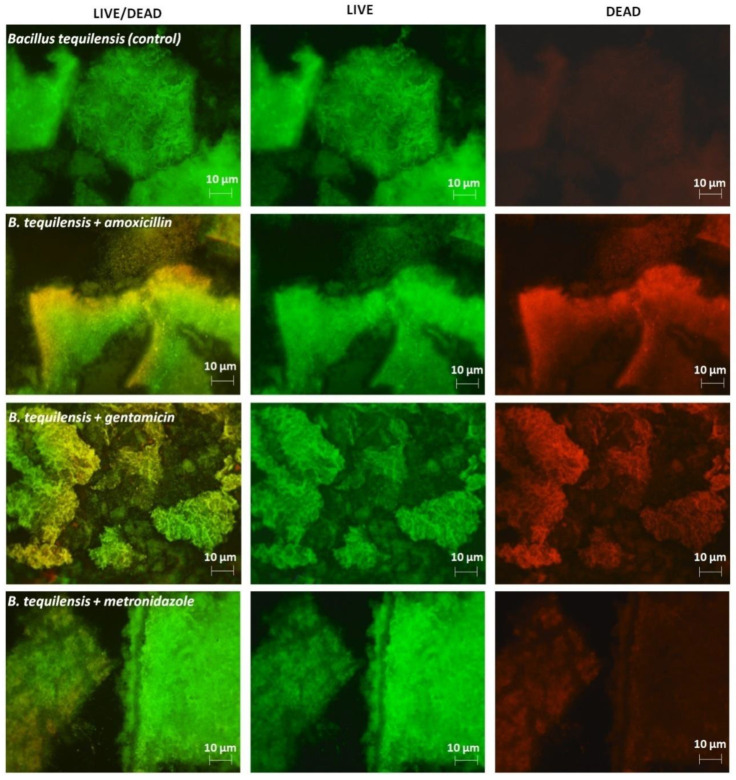
Fluorescence images of the biofilm formed by *B. tequilensis* bacteria treated with antibiotics, at magnifications of 100×.

**Figure 6 jcm-11-00722-f006:**
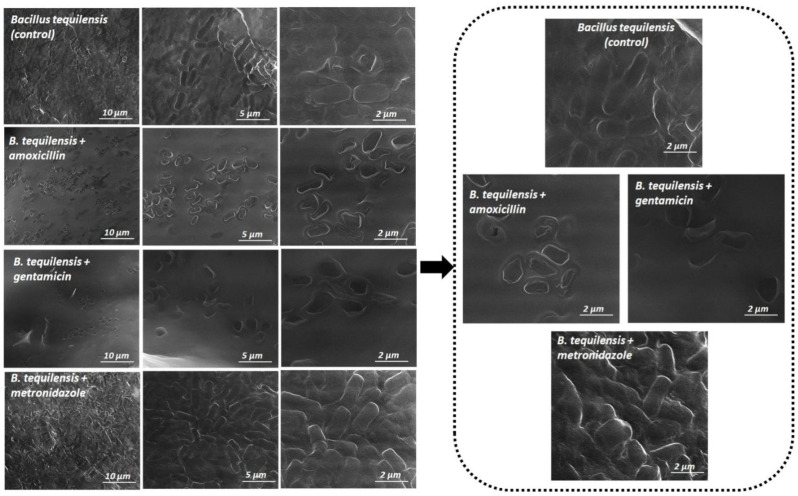
SEM images of the biofilm formed by *B. tequilensis* bacteria treated with antibiotics, before capillary electrophoresis at magnifications of 10.000×; 25.000× and 50.000× respectively.

**Figure 7 jcm-11-00722-f007:**
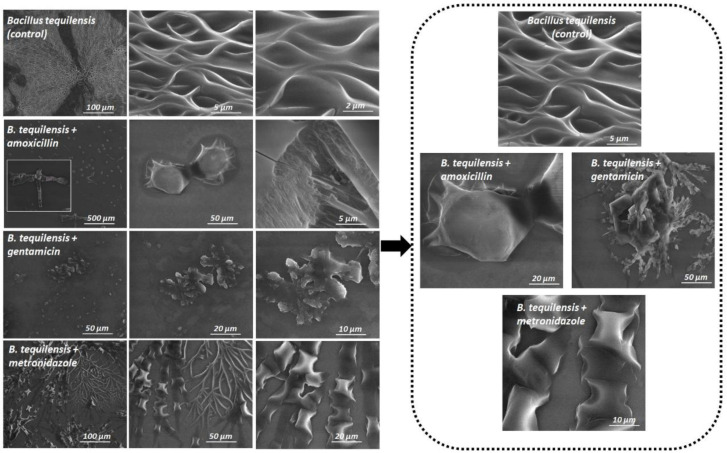
SEM images of the biofilm formed by *B. tequilensis* bacteria treated with antibiotics, after capillary electrophoresis, at magnifications of 1.000×; 2.500×; 10.000×; 25.000× and 50.000× respectively.

**Figure 8 jcm-11-00722-f008:**
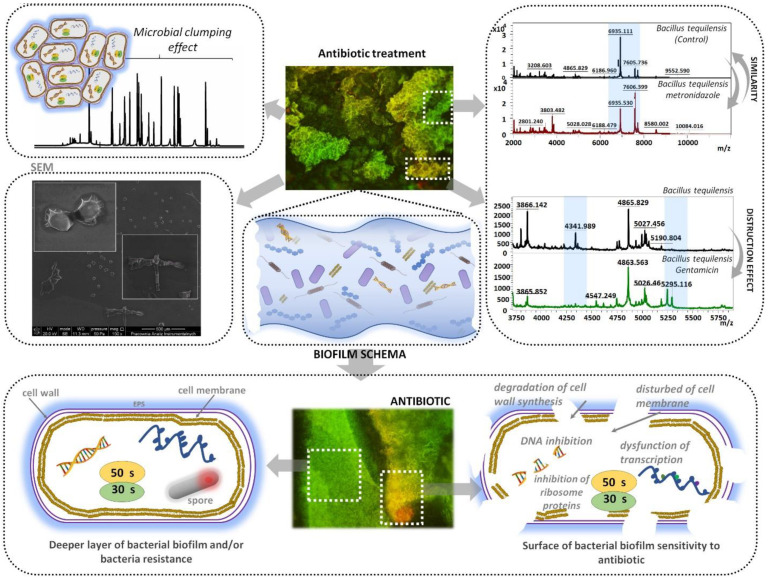
Scheme of mechanism of bacterial biofilm response—*B. tequilensis*.

## Data Availability

Not applicable.

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
