# Peer review of "Identification, Structure and Characterization of Bacillus tequilensis Biofilm with the Use of Electrophoresis and Complementary Approaches"

_jcm, 2022, doi:10.3390/jcm11030722_

Round 1

Reviewer 1 Report

Opinion on Pauter et al., “Identification, structure and characterization of Bacillus tequilensis biofilm with the use of electrophoresis and complementary approaches”

The authors apply capillary electrophoresis, zeta potential measurements, MALDI-TOF MS analysis, fluorescence microscopy and scanning electron microscopy to compare Bacillus tequilensis culture before and after treatment with three different antibiotics. Based on the observed differences, the authors draw various conclusions on the effect of these antibiotics.

I believe the authors attribute the loss of the slowest peak on the CE to the decrease of aggregation. Also, the observed increase in absolute Zeta potential could cause a lower tendency to aggregate.  I did not see any clear connection of these two physical parameters to changes in the MS profile, or morphology changes. Neither did I find any plausible explanation on why metronidazole had a somewhat different effect on the measured parameters than amoxicillin and gentamycin. I find these data, presented this way to be inappropriate for publication, for the reasons listed in detail below.

Major criticism:

  1. The objectives of this work are not clearly defined. The last paragraph of the introduction states: “Therefore, the main propose of this study was characterization of Bacillus tequilensis as an model of biofilm using capillary electrophoresis. (...)Moreover, we elucidated the role of bacteria biofilm ex-posure to antibiotics belonging to different therapeutic groups, with different spectrums and mechanisms of action, as well as various chemical structures, such as amoxicillin, gentamicin as well as metronidazole. ” Both, but especially the second sentence is difficult to comprehend. I assume the authors wished to study the EFFECTS of antibiotic administration on the biofilm formation of Bacillus tequilensis.

  1. In my opinion, the fundamental experimental setup is flawed. If the authors seek differences linked to biofilm formation, they should first compare planktonic and biofilm-derived Bacillus tequilensis cells. Or alternatively, a wild type bacterium with a gene knockout that has lost the ability of biofilm formation. I do not exclude the possibility that one or more of these three antibiotics inhibits biofilm formation at its MIC, but even if it does, we see a mixed effect on CE, ZP, MS spectrum and morphology: it is not dissected, which changes are attributable to cell physiology changes caused by the antibiotic, and which to the lack of biofilm formation. Also, I do not see any direct evidence that the biofilm formation was inhibited at this concentration, by either of the three antibiotics. Overall, if I had this task, I would first compare the CE, ZP, protein composition and morphology of planktonic and biofilm-forming bacteria. Second, when applying antibiotics on biofilm forming bacteria, and re-measuring the parameters, I would compare the results to the previous two datasets to see which changes are attributable to the loss or damage of the biofilm, and which changes are totally different, indicating that they are the results of cellular damage. I would also add a technique to quantitatively measure the secreted exopolysaccharide in all three cases (biofilm-forming, non-biofilm-forming, biofilm-forming +antibiotic).

  1. I do not see the methodology that warrants that the authors are investigating biofilms. As far as I can extract from the Methods, they are growing Bacillus tequilensis in liquid culture, then pelleting it by centrifugation. What fraction of these cells are planktonic, and what is within a biofilm? These are cornerstone parameters if one wants to study biofilm formation, and factors influencing it.

  1. It is not properly explained, how the authors obtained cell growth, if using antibiotics at the MIC concentration. Was there a significant amount of time allowing the evolution of resistant cells? Was cell growth quantitatively monitored at all?

Methods and Materials: This section requires a lot more attention to allow easy evaluation of results and repeatability of experiments. For example, the specific antibiotic concentrations used (0.5, 2 and 8 ug/ml): were these the actual MICs established for the specific strain used, or were these values obtained from EUCAST? The composition of the buffers (e.g. TB buffer, TBH buffer, etc.) is not provided. The rationale of the fluorescent microscopy method is not explained, neither here nor in the Results. This should be done at least briefly, in order to be able to evaluate the microscopic images (it is fine to refer to the cited article only concerning the details). I assume it is the uptake of a red dye by dead cells, but it is not explained. The green dye, and its function is not explained either. Repeating the SEM experiments would also require publishing a lot more details in the Methods. When comparing the SEM images of biofilms before and after CE, which peak of the CE was analyzed?

Capillary electrophoresis: electropherograms need more explanation. Which peaks mark cell components or debris, which mark individual cells, which mark clumps of cells? How is this verified?

I see that the slowest peak (seen at 12 min) is missing upon antibiotic treatment. The authors attribute this to the lack of the largest type of cell clump, and interpret this as successful inhibition of biofilm formation. But could it not be that the 12 min peak marks intact cells, seen without antibiotics, and this is missing under antibiotic treatment (in this case, all peaks coming earlier would mean cell debris of various sizes)? Also, we see a much smaller SUM(SAS) percentage for amoxicillin and gentamycin, What does this mean? Smaller biomass? If yes, was there any normalization for this difference when analyzing peak intensities? This should be explained.

About Zeta potentials: are the differences observed with and without antibiotics significant? The authors state that above +/-30 mV absolute ZP, the particles are stable in suspension (“Therefore, since the established zeta potential value (ξ = - 43.65mV) for the untreated bacterial cells is more than -30 mV, such system is considered stable [24].”). So are the observed differences biologically relevant at all?

MALDI-TOF:

The authors refer to multiple peaks without those peaks being annotated or visible on the figures. E.g. 6187 and 6435 in line 291; 6720 and 9878 in line 303, etc. These must be shown, if the authors wish to support something using them.

L 311: “In case of metronidazole the molecular profile was the most similar to the control, the result is from a weak action and mechanism of action inducing production of com-pounds inside cells that destroy microbial DNA [28-30].”

Why would DNA-breakdown result in an MS spectrum similar to that of the control? What do the authors mean with “weak action”? Aren’t all antibiotics administered at their MIC concentration, as mentioned in the Methods? In addition, references 28-30 do not say a single word about the mechanism of action of metronidazole.

Figure 4: What is depicted on the x and y axes? Is it the peak intensities corresponding to certain masses? So do the four charts show the correlation between the peak intensities measured under two conditions?  What do the bars and the lines show on the two axes? A proper explanation is missing from both the Methods and the figure legend. Was MALDI-TOF used for quantitative analysis? If yes, this has been shown to be very inaccurate, especially if no internal isotope control is used. Or was there an internal isotope control? What was it, and why was it not described? What rationale was used to choose the peaks for this analysis?

L 322: “First of all, downregulation of the proteins associated with the regu-lation of the protein translation and transcription as well as the proper structure of the ribosomes in amoxicillin and gentamicin treated cells were noted.” Please indicate in detail which genes the authors are talking about. Link the masses to gene names and gene function, preferably in a table, otherwise this statement is unsupported.

L 351: “No efficient antibiotics ac-tion will allowed the bacterial system to form biofilm.” I still don’t see the support for this statement. Which MS-markers were indicative of biofilm formation? What where the parameters describing “efficient antibiotics action”? Where do we see any connection (especially: an inverse correlation) between the two?

L 364. The authors describe metronidazole as  having weak bacteriocidal activity (MIC=8ug/ml), while the other two antibiotics as strong (2 and 0.5 ug/ml for amoxicillin and gentamycin, respectively). I understand, that gentamycin is the most POTENT of the three, followed by amoxicillin and the least potent metronidazole. But this is irrelevant, since all were applied at their MIC concentrations, weren’t they? What is relevant, is the EFFICACY of the three drugs. Are they different? If yes, what microbiological parameter was measured to conclude this?

L 374: “Furthermore, they evidenced that tobramycin at the biofilm periphery both stimulated a localized stress re-sponse and caused the death of bacteria cells those regions, but not in the deeper layers of the biofilm [33]. In our study, similar observation was noticed to B. tequilensis cells treated with gentamicin.” I do not see any figure comparing the morphology of cells from deep and superficial layers of the biofilm. 

Figure 5: A more detailed legend is required. E.g. what do the three columns represent?

L. 429: “The images obtained seem to reflect the proteomic changes noted during the MALDI analysis – mostly related to the disturb-ance of proper spores production.” I don’t see which MS signals and which morphology changes seen with SEM mark disturbance of spore production.

L 466: “According to MALDI profiles, the molecular changes are related to aggregates formation, visible by CE.” Which MS changes derive from aggregate formation? On the CE, the largest aggregate (giving a peak at 12) was said to be missing upon antibiotic treatment. Or are the authors suggesting the opposite (i.e. increased cell clumping or aggregation upon antibiotic administration?)?

Conclusions

L 488: Conclusions on Zeta potentials: “In the case of zeta potential measurements, the strong electrostatic interactions be-tween bacterial cells and used antibiotics were observed.” This obviously needs rephrasing (or did the authors really measure electrostatic interactions between cells and antibiotics?). I do not see what types of interactions were observed.

L 492. “MALDI- TOF MS analysis indicated changes in the molecular profiles of B. tequilensis before and after antibiotic therapy, leading to proposed mechanisms of antibiotic re-sistance.” It is true, that in two out of three antibiotic treatments, there were larger changes in the MS spectra. But these two were concerning the use of Amoxicillin and Gentamycin, which are two antibiotics with very different mechanisms of action. What molecular mechanisms explain resistance to both, strictly based on the observed data?

L 498. “According to microscopic and mass spectromet-ric analyses, it was notice that amoxicillin and gentamicin caused the degradation of cell wall synthesis, disturbed of the matrix of extracellular polysaccharide substances sur-rounding the biofilm (EPSs) and inhibition of ribosomal proteins and dysfunction of tran-scription, related to higher absolute value of ZP and less signals on CE.” The proper support of first part of sentence (MS indicating cell wall synthesis inhibition, ECM disturbance, translation and transcription inhibition) is missing, as mentioned earlier. If these details are given, the second part of the sentence still needs support: how do these changes seen in protein abundance explain the changes seen in CE experiments and ZP measurement?

Last, but not least, this manuscript needs lingual correction. Some sentences are misleading due to grammatical errors (e.g. “Molecular identification of bacteria biofilm formation” should be “Molecular identification of biofilm forming bacteria”)

Author Response

We would like to thank the Reviewers for careful reading, and constructive suggestions of our manuscript that will help us to improve our work. According to the comments from the reviewers, we comprehensively revised our manuscript. Hoping that we addressed all the questions mentioned by the reviewers, below we include the point by point response to each comment.

Reviewer(s)' Comments to the Author:

The authors apply capillary electrophoresis, zeta potential measurements, MALDI-TOF MS analysis, fluorescence microscopy and scanning electron microscopy to compare Bacillus tequilensis culture before and after treatment with three different antibiotics. Based on the observed differences, the authors draw various conclusions on the effect of these antibiotics.

I believe the authors attribute the loss of the slowest peak on the CE to the decrease of aggregation. Also, the observed increase in absolute Zeta potential could cause a lower tendency to aggregate.  I did not see any clear connection of these two physical parameters to changes in the MS profile, or morphology changes. Neither did I find any plausible explanation on why metronidazole had a somewhat different effect on the measured parameters than amoxicillin and gentamycin. I find these data, presented this way to be inappropriate for publication, for the reasons listed in detail below.

Major criticism:

  1. The objectives of this work are not clearly defined. The last paragraph of the introduction states: “Therefore, the main propose of this study was characterization of Bacillus tequilensis as an model of biofilm using capillary electrophoresis. (...) Moreover, we elucidated the role of bacteria biofilm ex-posure to antibiotics belonging to different therapeutic groups, with different spectrums and mechanisms of action, as well as various chemical structures, such as amoxicillin, gentamicin as well as metronidazole. ” Both, but especially the second sentence is difficult to comprehend. I assume the authors wished to study the EFFECTS of antibiotic administration on the biofilm formation of Bacillus tequilensis.

 Re: According to first sentence, the author specified the main goal of the study; means using of capillary electrophoresis (CE) technic as a tool for the biofilm characterization. Additionally, to prove the possibility of using of such technique in this direction, the authors set out to use complementary techniques that can confirm the observations from CE.

Concerning the second sentence, the authors means that in present study as a compounds that will change the surface of bacterial strain, in different way, were selected different antibiotics characterized by distinct action mechanism. In this context, three antibiotics with various chemical structures (aminoglycoside, nitroimidazole, β-lactam), distinct mechanisms of action (different molecular targets - inhibition of bacterial protein synthesis, disruption of cell wall biosynthesis, interaction with the DNA) as well as various MIC values (high action - gentamicin, medium - amoxicillin and low - metronidazole) have been chosen. In such way it was possible to change the electrophoretic mobility of the investigated biofilm, as a non-modified profiles of bacterial aggregates,  to monitor the possible use of the capillary electrophoresis in characterization of the modified biofilm. No effect study of antibiotics on the biofilm formation was planned, once the selected antibiotics are well known and are routinely use. The viability aspect have been also discussed due to the fact that it is well known that antibiotics can change the surface structure of the biofilm in different way depending on the action mechanism of antibiotics. Therefore, such antibiotics were deliberately chosen to modified the surface structure of the bacterial biofilm at different level.  

Moreover, studied antibiotic drugs belonging to different therapeutic groups are mostly applied among patients in the intensive care unit. Hence, from this point of view it was important for us to observe the potential influence of the relevant antibiotics on the formation and structure modification of the biofilm. On the other hand, antibiotic resistance has become life-threatening. A combination of used antibiotic drugs is commonly used as obligatory of SSI infections. Hence, the main aim of this study was to utilize an in vitro biofilm to test the effect of various antibiotics with different spectrum.

According to the Reviewer’s remarks the last section of the introduction was rewritten.

  1. In my opinion, the fundamental experimental setup is flawed. If the authors seek differences linked to biofilm formation, they should first compare planktonic and biofilm-derived Bacillus tequilensis cells. Or alternatively, a wild type bacterium with a gene knockout that has lost the ability of biofilm formation. I do not exclude the possibility that one or more of these three antibiotics inhibits biofilm formation at its MIC, but even if it does, we see a mixed effect on CE, ZP, MS spectrum and morphology: it is not dissected, which changes are attributable to cell physiology changes caused by the antibiotic, and which to the lack of biofilm formation. Also, I do not see any direct evidence that the biofilm formation was inhibited at this concentration, by either of the three antibiotics. Overall, if I had this task, I would first compare the CE, ZP, protein composition and morphology of planktonic and biofilm-forming bacteria. Second, when applying antibiotics on biofilm forming bacteria, and re-measuring the parameters, I would compare the results to the previous two datasets to see which changes are attributable to the loss or damage of the biofilm, and which changes are totally different, indicating that they are the results of cellular damage. I would also add a technique to quantitatively measure the secreted exopolysaccharide in all three cases (biofilm-forming, non-biofilm-forming, biofilm-forming +antibiotic).

 Re: We would like to thank the Reviewer for such valuable remarks. Indeed, simultaneous analysis of the both bacterial clumping (aggregates) and biofilm-derived cells (planktonic) could give additional information about impact of the investigated antibiotics on the biofilm formation, morphology, drugs modes of action against B. tequilensis. The same apply to the use of wild-type strains with loss of ability to biofilm formation. Such analysis should be included in further studies to obtain deepen picture of the noted phenomena in presented manuscript. Nevertheless, the aim of this study was to check the usefulness of the CE technique for the fast screening of the effect of the different antibiotics addition on the biofilm formation, as analysis of native (control) and treated electrophoretic profiles. The CE analysis is able to detect changes undergoing in the biofilm-derived B. tequilensis cells that enable tracking impact of the antibiotics on the biofilm development. Thus, to achieve this, we have chosen the B. tequilensis, belongs to the Bacillus sp. that are the model bacterial strains to biofilm formation since they can produce heat-resistant endospores that play an important role in bacterial persistence and biofilm formation as well as they are known to be able to produce extracellular polymeric substances (EPS) which play a key role in the resistance of biofilm to antibiotics by creating a mechanical barrier that restricts drug diffusion into bacterial cells (see K. P. Lemon, A. M. Earl, H. C. Vlamakis, C. Aguilar, and R. Kolter, Biofilm Development with an Emphasis on Bacillus subtilis. Curr Top Microbiol Immunol. 2008; 322: 1–16. doi: 10.1007/978-3-540-75418-3_1). The use of this strain under the incubation conditions established by us secured that vast majority of the cells form biofilm, therefore, influence of the planktonic cells on the results can be considered as irrelevant. Regarding EPS analysis, in the literature we can find studies focused on the deepen EPS production investigation concerning Bacillus species (e.g. Branda SS, Chu F, Kearns DB, Losick R, Kolter R. A major protein component of the Bacillus subtilis biofilm matrix. Mol Microbiol. 2006 Feb;59(4):1229-38. doi: 10.1111/j.1365-2958.2005.05020.x). Such studies are interesting and should be included during designing further studies, however, quantitatively measure the secreted exopolysaccharide are out of the main topic of the presented studies.

  1. I do not see the methodology that warrants that the authors are investigating biofilms. As far as I can extract from the Methods, they are growing Bacillus tequilensis in liquid culture, then pelleting it by centrifugation. What fraction of these cells are planktonic, and what is within a biofilm? These are cornerstone parameters if one wants to study biofilm formation, and factors influencing it.

Re: Thanks the Reviewer for suggestion. Since we used a model biofilm-forming bacteria, applied during experiment culture conditions ensured that the vast majority of the cells migrated to the air-liquid interface, where they formed a floating biofilm. Such phenomenon was indicated in the work Lemon et al. (2008). Therefore, chose of such strain enable us to reduce impact of the planktonic cells to the irrelevant level. The description of the applied culture conditions was supported with suitable information to clarify this issue – “A bacterial colony was grown on Tryptic Soy Agar (Sigma-Aldrich, India) during 24 h at 37°C to ensure that the vast majority of the cells migrated to the air-liquid interface, where they formed a floating biofilm”

  1. It is not properly explained, how the authors obtained cell growth, if using antibiotics at the MIC concentration. Was there a significant amount of time allowing the evolution of resistant cells? Was cell growth quantitatively monitored at all?

Re: Bacterial cells were cultured under standard microbiological conditions (TSA, 24h, 37 C).  Then, to modify the bacterial cell surface with antibiotics, bacterial cells were incubated with selected antibiotics (2h). The optimal incubation time (2h) has been applied based on our previous experiment already published and described by Buszewski, B et al [2019]. During the 2h cell surface modification is achieved; moreover, has been checked also the 24h of incubation but such long time destroyed totally the cells (in case of amoxicillin and gentamicin).  

*Buszewski, B.; Król, A.; Pomastowski, P.; Railean-Plugaru, V.; Szultka-MÅ‚yÅ„ska, M. Electrophoretic Determination of Lac-tococcus lactis Modified by Zinc Ions. Chromatographia 2019, 82, 347–355, doi:10.1007/s10337-018-3665-3.

Methods and Materials: This section requires a lot more attention to allow easy evaluation of results and repeatability of experiments. For example, the specific antibiotic concentrations used (0.5, 2 and 8 ug/ml): were these the actual MICs established for the specific strain used, or were these values obtained from EUCAST? The composition of the buffers (e.g. TB buffer, TBH buffer, etc.) is not provided. The rationale of the fluorescent microscopy method is not explained, neither here nor in the Results. This should be done at least briefly, in order to be able to evaluate the microscopic images (it is fine to refer to the cited article only concerning the details). I assume it is the uptake of a red dye by dead cells, but it is not explained. The green dye, and its function is not explained either. Repeating the SEM experiments would also require publishing a lot more details in the Methods. When comparing the SEM images of biofilms before and after CE, which peak of the CE was analyzed?

 Re: It has been chosen three antibiotics with different chemical structures (aminoglycoside, nitroimidazole, β-lactam), distinct mechanisms of action (different molecular targets - inhibition of bacterial protein synthesis, disruption of cell wall biosynthesis, interaction with the DNA) as well as various MIC values (high action - gentamicin, medium - amoxicillin and low - metronidazole) in order to change the biofilm surface structure in different way. Three antibiotics that demonstrated various modes of action were selected knowing, that they may be characterized also by the different ability to EPS penetration - a key factor in determining the effectiveness of an antibiotic in biofilm eradication. The MIC value was chosen according to the EUCAST guidelines characteristic for Bacillus sp.

Buffer solution composition has been completed in revised version of manuscript according to methods described in our previous work (Dziubakiewicz, E.; Buszewski, B. Capillary electrophoresis of microbial aggregates. Electrophoresis 2014, 35, 1160–1164, doi:10.1002/elps.201300588).

The fluorescence microscope (FM) was applied in order to visualized the level of antibiotic effect in context of the viability of the biofilm-derived bacterial cells. The method used for the respective experiment was previously described by our group using the ethidium bromide and acridine orange as a dyes in order to differentiate the live and dead cells to be able to establish the level of the modified biofilm structure.

The SEM experiment was performed in simple way by applying a small drop of bacteria cells suspension on the dedicate microscope slide. Then, the specimens were completely dried and directly examined by scanning electron microscopy (SEM).

During the interpretation of CE results (before and after), transfer and represented this aspect by SEM, all signals registered  as electrophoretic profiles of whole bacterial aggregates were taken in consideration.

According to Reviewer’s remarks and comments all aspect have been taken into account and the respective sections have been rewrite in order to avoid any misunderstanding.

Capillary electrophoresis: electropherograms need more explanation. Which peaks mark cell components or debris, which mark individual cells, which mark clumps of cells? How is this verified?

Re:  In the present work have been analyzed sum of the surface area of all signals. In our study, the applied capillary electrophoresis allowed to understand the total changes occurring in the cell membrane of biofilm-forming bacteria under the exposure to antibiotics in context of the clumping phenomenon (changes in electrophoretic intensities and disappearance of large aggregates).In present research the electrophoretic profiles of bacterial aggregates of biofilm was studied. After validation procedure, focused on reproducibility of electropherograms, whole  bacterial aggregates was collected (several times). Fractionation of bacterial aggregates, “from separated peak”, from technical point of view on our electrophoretic system is not possible; moreover in context of viability, how was represented in our work via SEM and FM techniques. To prove and indicate which signal is responsible for debris, individual cells even in control sample it is very encouraging, especially in context of such complex matrix as biofilm. As we can observed from the SEM figure after electrophoresis (in case of antibiotic treatment) we can observe not only bacterial cells but also fragments with different shapes (namely debris), that will have different mobility, consequently can induce erroneous interpretation. Moreover, as we can see from SEM and FM pictures in case of control sample it is impossible to visualized separated cells or debris, even after electrophoresis. Unequivocal, if we would like to prove some hypothesis we have always take into account the control sample, then we can formulate conclusion.    

I see that the slowest peak (seen at 12 min) is missing upon antibiotic treatment. The authors attribute this to the lack of the largest type of cell clump, and interpret this as successful inhibition of biofilm formation. But could it not be that the 12 min peak marks intact cells, seen without antibiotics, and this is missing under antibiotic treatment (in this case, all peaks coming earlier would mean cell debris of various sizes)? Also, we see a much smaller SUM(SAS) percentage for amoxicillin and gentamycin, What does this mean? Smaller biomass? If yes, was there any normalization for this difference when analyzing peak intensities? This should be explained.

Re: Based on electrokinetic theories related to pseudoisotachophoretic modes (Dziubakiewicz, E.; Buszewski, B. Capillary electrophoresis of microbial aggregates. Electrophoresis 2014, 35, 1160–1164, doi:10.1002/elps.201300588; Rogowska, A.; Pomastowski, P.; ZÅ‚och, M.; Railean-Plugaru, V.; Król, A.; RafiÅ„ska, K.; Szultka-MÅ‚yÅ„ska, M.; Buszewski, B. The influence of different pH on the electrophoretic behaviour of Saccharomyces cerevisiae modified by calcium ions. Scientific Reports 2018, 8, 2–11, doi:10.1038/s41598-018-25024-4), all of the signal of analyzed bacteria sample registered on electropherograms represents the bacterial aggregates. Therefore, the signal registered at 12 min will be considered responsible for the clumping having high tendency to represent larger aggregates. On another hands, considered the untreated bacterial biofilm, all signals registered in electrophoregram marks intact cells just with different size. This aspect it was also proven by SEM and FM techniques. The signal registered at 12 min in control samples that is responsible for large aggregates disappears in the samples treated with antibiotics due to the fact that antibiotic in their specific way (action mechanism) divide the biofilm in different fragments with the smaller size (as it is proven by the SEM). Moreover, this phenomenon was also proven by increasing of zeta potential value in the samples treated with antibiotics compared to the control.     

About Zeta potentials: are the differences observed with and without antibiotics significant? The authors state that above +/-30 mV absolute ZP, the particles are stable in suspension (“Therefore, since the established zeta potential value (ξ = - 43.65mV) for the untreated bacterial cells is more than -30 mV, such system is considered stable [24].”). So are the observed differences biologically relevant at all?

Re: The zeta potential value is a parameter that will indicate the dispersion stability of system. In present study all investigated systems were found to be dispersive stable. However, in our case the increasing of the zeta potential value after antibiotic treatment may be caused by degradation of bacterial cells into smaller micellar system determined by higher solvation effects, increase the dispersion of the analyzed system. Therefore, in case of present study the ZP value indicate the differences after antibiotic treatment.

MALDI-TOF:

The authors refer to multiple peaks without those peaks being annotated or visible on the figures. E.g. 6187 and 6435 in line 291; 6720 and 9878 in line 303, etc. These must be shown, if the authors wish to support something using them.

Re: All signals have been discussed in the text, to represent all characteristic signals in one figure is difficult, however, the figure was improved and proposed. We think  that Reviewer will find it justified. 

L 311: “In case of metronidazole the molecular profile was the most similar to the control, the result is from a weak action and mechanism of action inducing production of com-pounds inside cells that destroy microbial DNA [28-30].”

Why would DNA-breakdown result in an MS spectrum similar to that of the control? What do the authors mean with “weak action”? Aren’t all antibiotics administered at their MIC concentration, as mentioned in the Methods? In addition, references 28-30 do not say a single word about the mechanism of action of metronidazole.

Re: Thanks the Reviewer for insightful remark. We agree that mentioned sentence was written in unclear way, therefore, we rewrite it to clear emphasize meaning of the results – “In case of metronidazole the molecular profile was the most similar to the control indicating limiting entering antibiotic molecules into the cells and therefore, missing access to their molecular target which is the microbial DNA”. Regarding references, they refer also to earlier results. We want to apologies for this mistake and in the revised version of the manuscript we combined paragraphs describing all three antibiotics to avoid further misunderstanding.  

Figure 4: What is depicted on the x and y axes? Is it the peak intensities corresponding to certain masses? So do the four charts show the correlation between the peak intensities measured under two conditions?  What do the bars and the lines show on the two axes? A proper explanation is missing from both the Methods and the figure legend. Was MALDI-TOF used for quantitative analysis? If yes, this has been shown to be very inaccurate, especially if no internal isotope control is used. Or was there an internal isotope control? What was it, and why was it not described? What rationale was used to choose the peaks for this analysis?

Re: We want to thank the Reviewer for this remark. Figure 4 shows comparison of signals appearing noticed on MALDI-TOF mass spectra for control Bacillus tequilensis cells (x axis) and under antibiotics treatment (y axis) which were proteins (marked with different numbers #1 -#10) identified using UNIPROT database – proteins are listed in the figure captions with their m/z values. Details description of the compared proteins with references are provided in main text in paragraph “3.4. MALDI- TOF MS analysis”. Values presented in the x and y axis are not intensities of the signals but m/z values – we used histogram not correlation analysis in order to show differences in distribution of the signals between treated and untreated B. tequilensis cells.

L 322: “First of all, downregulation of the proteins associated with the regu-lation of the protein translation and transcription as well as the proper structure of the ribosomes in amoxicillin and gentamicin treated cells were noted.” Please indicate in detail which genes the authors are talking about. Link the masses to gene names and gene function, preferably in a table, otherwise this statement is unsupported.

 Re: According to the Reviewer comments the description of the results has been supplied with the suitable information. “First of all, downregulation of the proteins associated with the regulation of the protein translation and transcription as well as the proper structure of the ribosomes in amoxicillin and gentamicin treated cells were noted: 6720 m/z - 50S ribosomal protein L32 (structural constituent of ribosome; Uniprot,[34]), 9552 m/z- Protein Veg (regulation of transcription, DNA-templated; Uniprot, [33]).

L 351: “No efficient antibiotics action will allowed the bacterial system to form biofilm.” I still don’t see the support for this statement. Which MS-markers were indicative of biofilm formation? What where the parameters describing “efficient antibiotics action”? Where do we see any connection (especially: an inverse correlation) between the two?

 Re: In order to avoid any further misunderstanding the authors revised the respective section and the unclear affirmation was excluded. Thank you the Reviewer for the Remark.

L 364. The authors describe metronidazole as having weak bacteriocidal activity (MIC=8ug/ml), while the other two antibiotics as strong (2 and 0.5 ug/ml for amoxicillin and gentamycin, respectively). I understand, that gentamycin is the most POTENT of the three, followed by amoxicillin and the least potent metronidazole. But this is irrelevant, since all were applied at their MIC concentrations, weren’t they? What is relevant, is the EFFICACY of the three drugs. Are they different? If yes, what microbiological parameter was measured to conclude this?

 Re: In our study, we have chosen three antibiotics with different chemical structures (aminoglycoside, nitroimidazole, β-lactam), distinct mechanisms of action (different molecular targets - inhibition of bacterial protein synthesis, disruption of cell wall biosynthesis, interaction with the DNA) as well as various MIC values (high action - gentamicin, medium - amoxicillin and low - metronidazole). The relevance of present study is fact that based on the well-known knowledge regarding the respective antibiotic action it was possible to change the biofilm surface structure in different way. In such way it was possible to change the electrophoretic mobility of the investigated biofilm to monitor the possible use of the capillary electrophoresis in characterization of the modified biofilm.

Based on the reviewer’s remarks the manuscript has been supplemented with an additional explanation in order to allowed the reader to follow the idea.

L 374: “Furthermore, they evidenced that tobramycin at the biofilm periphery both stimulated a localized stress response and caused the death of bacteria cells those regions, but not in the deeper layers of the biofilm [33]. In our study, similar observation was noticed to B. tequilensis cells treated with gentamicin.” I do not see any figure comparing the morphology of cells from deep and superficial layers of the biofilm.

 Re: According to the Figure 5 made by fluorescence microscopy visualizing the viability of the investigated biofilm, it is possible to distinguish the more or less affected zone in dependence of the color (green – alive, red – dead or affected). The structure of the biofilm is affected in different way  (deep or superficial layers) based on the used antibiotics. More evidence effect are represented by the SEM images (before and after electrophoresis).

Considering the Reviewer’s remark the section was revised.

Figure 5: A more detailed legend is required. E.g. what do the three columns represent?

Re: the figure have been improved as was suggested by Reviewer.

  1. 429: “The images obtained seem to reflect the proteomic changes noted during the MALDI analysis – mostly related to the disturbance of proper spores production.” I don’t see which MS signals and which morphology changes seen with SEM mark disturbance of spore production.

 Re: Thanks the Reviewer for important comment. Regarding MS signals, disturbance of the spore production is mainly marked by lack of the signal 9878 m/z in the B. tequilensis cells treated with gentamicin and amoxicillin. This signal represent protein YwcE which is required for proper spore morphogenesis and germination. SEM images from amoxicillin and gentamicin variants supported these findings since disrupted cells or cells with distorted cell wall morphology were not accompanied by presence of spore-like structures which should appear after cell envelope destroying when sporulation process is not disturbed.

L 466: “According to MALDI profiles, the molecular changes are related to aggregates formation, visible by CE.” Which MS changes derive from aggregate formation? On the CE, the largest aggregate (giving a peak at 12) was said to be missing upon antibiotic treatment. Or are the authors suggesting the opposite (i.e. increased cell clumping or aggregation upon antibiotic administration?)?

Re: the hypothesis has been made based on the m/z changes recorded by MALDI technique for each sample separately (described in MALDI section). The author means the molecular changes characteristic for the each antibiotics. According to the Reviewer’s remark the sentence was rewritten.  

Conclusions

L 488: Conclusions on Zeta potentials: “In the case of zeta potential measurements, the strong electrostatic interactions be-tween bacterial cells and used antibiotics were observed.” This obviously needs rephrasing (or did the authors really measure electrostatic interactions between cells and antibiotics?). I do not see what types of interactions were observed.

 Re: A such hypothesis has been made based on the obtained results and interpretation described in details in the text. The presence of an electrical charge on the surface of microorganisms consequently has a direct impact on their aggregation, and can also serve as a parameter to distinguish between different bacterial strains. The electrical properties of bacterial cells can therefore be characterized by measuring the zeta potential, as one of electrokinetic parameters related to dispersion stability and solvation phenomena. We noticed that the value of zeta potential increases, therefore, the electrostatic interactions between bacterial cells and antibiotics will be stronger, the stability of the systems will increase, and the size distribution will be more homogeneous. Moreover, bacteria are endowed with negative surface charge in studied condition. According to pKa values of most impact surface functional groups of bacterial system (treated as biocolloids). This charge determines the formation of electric double layer and consequently zeta potential. These parameters, in turn, are closely related to the adhesion and colonization ability of bacteria to various surfaces, including tissues. We used the zeta potential measured to estimate the surface charge of bacterial system by measuring the zeta potential. Moreover, zeta potential is related to electrophoretic mobility studied by CE approaches. Thus, it is possible to explain the phenomenon of bacterial cell aggregation during electrophoresis and strong adhesion to the surface of capillary surfaces

 In order to avoid misunderstanding the sentences was rewritten.

L 492. “MALDI- TOF MS analysis indicated changes in the molecular profiles of B. tequilensis before and after antibiotic therapy, leading to proposed mechanisms of antibiotic re-sistance.” It is true, that in two out of three antibiotic treatments, there were larger changes in the MS spectra. But these two were concerning the use of Amoxicillin and Gentamycin, which are two antibiotics with very different mechanisms of action. What molecular mechanisms explain resistance to both, strictly based on the observed data?

Re: We would like to thank the Reviewer for the important observation. It has been  chosen three antibiotics with different chemical structures (aminoglycoside - gentamicin, nitroimidazole - metronidazole, β-lactam - amoxicillin), distinct mechanisms of action (different molecular targets - inhibition of bacterial protein synthesis [gentamicin], disruption of cell wall biosynthesis [amoxicillin], interaction with the DNA [metronidazole]) as well as various MIC values (high action - gentamicin, medium - amoxicillin and low - metronidazole) in order to change the biofilm surface structure in different way. Three antibiotics that demonstrated various modes of action were selected knowing, that they may be characterized also by the different ability to EPS penetration - a key factor in determining the effectiveness of an antibiotic in biofilm eradication. Suitable information explaining the antibiotics choose has been added to the revised version of the manuscript.

Regarding observed data that can decipher potential molecular mechanisms involved in the biofilm development disturbance both SEM images, CE analysis as well as results of the proteins profiles changes indicated significant differences between investigated antibiotics in terms of cell morphology, presence or lack specific proteins related to spore production, proteins transcription and regulation what reflected in different electrophoregrams images.      

L 498. “According to microscopic and mass spectrometric analyses, it was notice that amoxicillin and gentamicin caused the degradation of cell wall synthesis, disturbed of the matrix of extracellular polysaccharide substances sur-rounding the biofilm (EPSs) and inhibition of ribosomal proteins and dysfunction of tran-scription, related to higher absolute value of ZP and less signals on CE.” The proper support of first part of sentence (MS indicating cell wall synthesis inhibition, ECM disturbance, translation and transcription inhibition) is missing, as mentioned earlier. If these details are given, the second part of the sentence still needs support: how do these changes seen in protein abundance explain the changes seen in CE experiments and ZP measurement?

 Re: The hypothesis has been formulated based on the identified signals by UNIPROT database and it is attend to express the correlation of the MALDI, CE results and ZP. However, in order to avoid any inconvenience the senesce was revised and improved.    

Last, but not least, this manuscript needs lingual correction. Some sentences are misleading due to grammatical errors (e.g. “Molecular identification of bacteria biofilm formation” should be “Molecular identification of biofilm forming bacteria”)

Re: The gramma in whole manuscript has been checked by English native speaker.

Thank you very much for your critical review. It was very useful in the correction of our manuscript. Identification of weak points throughout the text helped us to increase the value of our paper. Most of the comments and changes suggested by Reviewer have been incorporated into the manuscript. The rest one are inspiration for us for the future experiments. Once again, thank you very much for your help.

Reviewer 2 Report

Dear authors

It was pleasure to review this important manuscript, however i have some minor concerns like only one antibiotic concentration was used which needs more elaboration why this concentration only was chosen. I was disappointed with the SEM study because more fields are needed to elucidate and confirm the antibiofilm activity. Lastly, i would like to know why B. tequilensis was chosen specifically it would be more sounded if you chose a pathogenic strain that cause CA-UTI as an example.

Author Response

We would like to thank the Reviewers for careful reading, and constructive suggestions for our manuscript that will help us to improve our work. According to the comments from the reviewers, we comprehensively revised our manuscript. Hoping that we addressed all the questions mentioned by the reviewers, below we include the point-to-point response to each comment.

In the manuscript file all the changes have been provided with  using the “Track Changes” function.

Reviewer(s)' Comments to the Author:

It was pleasure to review this important manuscript, however I have some minor concerns like only one antibiotic concentration was used which needs more elaboration why this concentration only was chosen. I was disappointed with the SEM study because more fields are needed to elucidate and confirm the antibiofilm activity. Lastly, i would like to know why B. tequilensis was chosen specifically it would be more sounded if you chose a pathogenic strain that cause CA-UTI as an example.

Answer: We would like to thank the Reviewer for the important comments.

We are agree with the Reviewer that choosing an pathogen strain can elucidate the antibiotic effect at clinical level. However, once the main goal of the study was to use Capillary electrophoresis as a tool for characterization of the biofilm it has been chosen  an safe Bacillus tequilensis strain as a model. We have chosen the B. tequilensis, because the Bacillus sp. are the model bacterial strains to biofilm formation. Bacillus species can produce heat-resistant endospores that play an important role in bacterial persistence and biofilm formation. In addition, these bacteria also possess swarm motility, which can facilitate microbial survival in the environment and colonization of surfaces, leading to biofilm formation. However, based on the previous literature the antibacterial effect of  various Bacillus spp. have been studied however  the B. tequilensis has not yet significant reported.  Moreover, in order to monitor how the  electrophoretic mobility is changed under stress condition, three antibiotics based on their different mechanism action were selected. In this context, it was enough only one concentration to achieve this scope  and to monitor the bacteria clumping in electric field under different antibiotic treatment. Obviously, the using of several concentrations to identified the electrophoretic mobility and how different concentrations can influence the migration aspect is unequivocal and of course we will take this aspect in consideration in the future; moreover, on pathogenic strains.

Regarding the SEM analysis, we are agree that to elucidate the antimicrobial effect of several antibiotics it is required more field to performed. However, in present research it was used SEM technique in order to prove the  Capillary electrophoresis results and how different antibiotics with different action mechanism can influence the migration of the investigated bacterial cell in electric field.

Reviewer 3 Report

Manuscript details:

Journal: Journal of Clinical Medicine

Manuscript ID: jcm-1524969

Title: Identification, structure and characterization of Bacillus tequilensis biofilm with the use of electrophoresis and complementary approaches

General Comments: The author explored to design a complementary approach in biofilm characterization before and after antibiotic treatment. 16S rRNA gene sequencing allowed for the identification of Bacillus tequilensis, as a microbial model of the biofilm formation. Also demonstrate the capability to characterize and show the differences of the electrophoretic mobility between untreated and treated biofilm with antibiotics: amoxicillin, gentamicin, metroni-dazole.

This reviewer perused this manuscript with great interest and pleasure, and truly believes in its scientific contribution to the field of Science & Technology. They also pointed out degradation/remodeling of the EPSs matrix, inhibition of cell wall synthesis and blocking the ribosomal protein synthesis by amoxicillin and gentamicin. However, this reviewer enlisted some observation regarding the article.

First and foremost: The manuscript is well-written, with particular attention to English spelling, grammar, syntax, and semantics, as well as scientific style. The authors should correct the remarks below before accepting the manuscript by editors.                                    

Specific comments:

Comment 1: In line 20, add ‘and’ prior to metronidazole.

Comment 2: In line 25-26, However, ….of the system. Which system please add.

Comment 3: In line 33, “range of strategies”- explain about it.

Comment 4: In line 35, “multilayer system of bacteria cells”- what do you mean by system?

Comment 5: In line 43-44, “determines significant resistance”- meaning is not clear. Rewrite the sentence.

Comment 6: In line 56, Remove comma after “Escherichia coli”.

Comment 7: In line 58, Use “Ca2+” instead of elaboration.

Comment 8: In line 67-68, …….., respectively. Please add reference after end of sentence.

Comment 9: In line 79, Use “complex” instead of prominent.

Comment 10: In line 85, a model.

Comment 11: In line 87, Remove “by CE”.

Comment 12: Line 98-99, Please revise the line (according to the previous protocol).

Comment 13: Line 99, Mention about the isolation source and specificity.

Comment 14: Line 106, What was the reason to choose only three antibiotics? Author should check more antibiotics from different classes.

Comment 15: Line 118, 13,000 rpm.

Comment 16: Line 131, Elaborate the “CZE” first time.

Comment 17: Line 155, “as a the average”- make correction.

Comment 18: Line 157, B. tequilensis.

Comment 19: Line 160, 14,400 rpm.

Comment 20: Line 165, 13,000 rpm.

Comment 21: Line 180, Explain the SEM procedure and use appropriate reference.

Comment 22: Line 184, Surface used for biofilm formation is missing. Please mention about surface.

Comment 23: Line 191 and 200, B. tequilensis.

Comment 24: In line 211, Please remove ‘s’ from controls.

Comment 25: Line 218, Remove comma.

Comment 26: In line 244, Please add “and” between (peptidoglycan, proteins) and removed the extra space after proteins (proteins ).

Comment 27: Please in Figure 3, 4, and 5 make italic B. tequilensis

Comment 28: In line 415 and 417, What was the reason to use different magnification range in SEM? Using the same magnification range in control and treatment group is recommended.

Introduction: you may add some relevant paper about biofilm and EPS. https://doi.org/10.1016/j.marpolbul.2021.112927, https://doi.org/10.1016/j.foodcont.2021.108179, https://doi.org/10.1016/j.psj.2021.101209

Author Response

We would like to thank the Reviewers for careful reading, and constructive suggestions for our manuscript that will help us to improve our work. According to the comments from the reviewers, we comprehensively revised our manuscript. Hoping that we addressed all the questions mentioned by the reviewers, below we include the point-to-point response to each comment.

In the manuscript file all the changes have been provided with  using the “Track Changes” function.

Reviewer(s)' Comments to the Author:

The author explored to design a complementary approach in biofilm characterization before and after antibiotic treatment. 16S rRNA gene sequencing allowed for the identification of Bacillus tequilensis, as a microbial model of the biofilm formation. Also demonstrate the capability to characterize and show the differences of the electrophoretic mobility between untreated and treated biofilm with antibiotics: amoxicillin, gentamicin, metronidazole.

This reviewer perused this manuscript with great interest and pleasure, and truly believes in its scientific contribution to the field of Science & Technology. They also pointed out degradation/remodeling of the EPSs matrix, inhibition of cell wall synthesis and blocking the ribosomal protein synthesis by amoxicillin and gentamicin. However, this reviewer enlisted some observation regarding the article.

First and foremost: The manuscript is well-written, with particular attention to English spelling, grammar, syntax, and semantics, as well as scientific style. The authors should correct the remarks below before accepting the manuscript by editors.  

Re: We would like to thank the Reviewer for the important comments.

The comments 1-3, 6-13 and 15-27  have been taken in full consideration and the respective sections have been revised and improved.

Comment 4: In line 35, “multilayer system of bacteria cells”- what do you mean by system?

Re: Thanks the Reviewer for the important comment. We suggested that the bacterial cells are able to accumulation and create the multi-layers. In order to avoided the misunderstanding the sentence in the manuscript has been revisited.

Comment 5: In line 43-44, “determines significant resistance”- meaning is not clear. Rewrite the sentence.

Re: Thanks for the Reviewer valuable remark. The significant resistance means that the most of the bacterial cells that create a biofilm demonstrate resistance to antibiotics. According to the Reviewer’s remarque the sentence was revised.

Comment 14: Line 106, What was the reason to choose only three antibiotics? Author should check more antibiotics from different classes.

Re: We would like to thank the Reviewer for the important observation. In our study, we have chosen three antibiotics with different chemical structures (aminoglycoside, nitroimidazole, β-lactam), distinct mechanisms of action (different molecular targets - inhibition of bacterial protein synthesis, disruption of cell wall biosynthesis, interaction with the DNA) as well as various MIC values (high action - gentamicin, medium - amoxicillin and low - metronidazole) in order to change the biofilm surface structure in different way. In such way it was possible to change the electrophoretic mobility of the investigated biofilm to monitor the possible use of the capillary electrophoresis in characterization of the modified biofilm.  

Comment 28: In line 415 and 417, What was the reason to use different magnification range in SEM? Using the same magnification range in control and treatment group is recommended.

Re: Thanks the Reviewer for the important comment. We are agree with the Reviewer, when is investigate the antimicrobial activity of the selected antibiotics and want to compare their action on the same bacterial strain it is required the same magnification. Once we used SEM technique only to point out the significance changes in the bacteria cells surfaces under various antibiotics treatment have selected different magnification just to show more clearly the modification and in such way to prove the electrophoresis results.

Introduction: you may add some relevant paper about biofilm and EPS. https://doi.org/10.1016/j.marpolbul.2021.112927, https://doi.org/10.1016/j.foodcont.2021.108179, https://doi.org/10.1016/j.psj.2021.101209

Re: According to Reviewer’s suggestion we added the references in the text.
